# Outcomes of Surgical Treatment for Extradural Benign Primary Spinal Tumors in Patients Younger than 25 Years: An Ambispective International Multicenter Study

**DOI:** 10.3390/cancers15030650

**Published:** 2023-01-20

**Authors:** Alexander C. Disch, Stefano Boriani, Aron Lazary, Laurence D. Rhines, Alessandro Luzzati, Ziya L. Gokaslan, Charles G. Fisher, Michael G. Fehlings, Michelle J. Clarke, Dean Chou, Nicole M. Germscheid, Klaus-Dieter Schaser, Jeremy J. Reynolds

**Affiliations:** 1University Center for Orthopedics, Trauma Surgery and Plastic Surgery, University Comprehensive Spine Center, University Hospital Carl Gustav Carus Dresden at the TU Dresden, 01307 Dresden, Germany; 2I.R.C.C.S. Istituto Ortopedico Galeazzi, 20161 Milan, Italy; 3National Center for Spinal Disorders, 1126 Budapest, Hungary; 4Department of Neurosurgery, MD Anderson Cancer Center, Houston, TX 77030, USA; 5Department of Neurosurgery, The Warren Alpert Medical School of Brown University, Rhode Island Hospital and The Miriam Hospital, Providence, RI 02903, USA; 6Department of Orthopaedics, Faculty of Medicine, The University of British Columbia, Vancouver, BC V5Z 1M9, Canada; 7Department of Surgery Halbert Chair, Spinal Program University of Toronto, Toronto Western Hospital University Health Network, Toronto, ON M5T 2S8, Canada; 8Department of Neurosurgery, Mayo Clinic, Rochester, MN 55902, USA; 9Department of Neurosurgery, The UCSF Spine Center, University of California, San Francisco, CA 94143, USA; 10AO Spine Knowledge Forum Tumor, AO Spine, 7270 Davos, Switzerland; 11Oxford Spinal Surgery Unit, Oxford University Hospitals, Oxford OX3 7LE, UK

**Keywords:** benign primary tumor, spinal tumor, adolescent, children, aggressive resection, surgical outcome

## Abstract

**Simple Summary:**

Compared to secondary lesions, primary spinal tumors are rare. Moreover, in extradural benign primary tumors, surgery is not always necessary, and oncological management expertise is limited even in spine centers. The following study aims to provide descriptive data on the effect of different resection strategies on local recurrence, and survival in patients suffering from benign primary spinal tumors younger than 25 years of age from an ambispective cross-sectional follow-up (PTRetro) study performed by the AO Spine Knowledge Forum Tumor. Compared to results previously published by this group, the aforementioned younger patient cohort presented without a correlation between the grade of aggressiveness in resection and local recurrence rates.

**Abstract:**

Extradural primary spinal tumors were retrospectively analyzed from a prospective database of 1495 cases. All subjects with benign primary tumors under the age of 25 years, who were enrolled between 1990 and 2012 (Median FU was 2.4 years), were identified. Patient- and case-related characteristics were collected and statistically analyzed. Results: 161 patients (66f;95m; age 17.0 ± 4.7 years at time of diagnosis) were identified. The most common tumors were osteoblastomas *n* = 53 (32.9%), osteoid osteomas *n* = 45 (28.0%), and aneurysmal bone cysts *n* = 32 (19.9%). The tumor grade, according to the Enneking Classification S1/S2/S3, was 14/73/74 (8.7/45.3/46.0%), respectively. Tumor-related pain was present in 156 (96.9%) patients. Diagnosis was achieved by biopsies in 2/3 of the cases. Spinal fixation was used in >50% of the cases. Resection was Enneking appropriate in *n* = 100 (62.1%) of cases. Local recurrence occurred in 21 (13.1%) patients. Two patients died within a 10-year follow-up period. Conclusion: This is one of the largest international multicenter cohorts of young patients surgically treated for benign spinal tumors. The heterogenic young patient cohort presented at a mid-term follow-up without a correlation between the grade of aggressiveness in resection and local recurrence rates. Further prospective data are required to identify prognostic factors that determine oncological and functional outcomes for young patients suffering from these rare tumors.

## 1. Introduction

Compared to secondary spinal lesions, primary benign tumors are a rare condition and present with a heterogeneous range of biological activity [1,2,3,4,5]. According to the widely accepted Enneking staging system for primary bone tumors [6,7,8], benign entities significantly vary from being classified as “latent (S1)” with slow intracompartmental growth, “active (S2)” defined as clinically symptomatic but still intracompartmental, and “aggressive (S3)”, characterized as non-respecting of compartmental borders and, as an extreme, present as tumors with malignant-like behavior including the ability to metastasize [6,8].

Oncological approaches for primary spinal tumors in younger patients often significantly differ from adult treatment regimens. In the common literature they are thereby divided into two age groups: from birth until 14 years of age, and from 15 to 25 years of age [3,9]. However, multidisciplinary adjuvant treatment often recommends similar treatment protocols for the same entities for all the patients under 25 years of age [1]. While internationally coordinated adjuvant treatment algorithms and study protocols are used, knowledge about onco-surgical outcomes of extradural benign primary spinal tumors in young patients is limited. Neither unique resection strategies (e.g., Enneking Inappropriate (EI) vs. Enneking Appropriate (EA)) nor clear-cut reconstruction guidelines exist.

The necessity for surgical treatment is closely related to underlying biological activity—graded by existing symptoms—the radiological appearance and staging, as well as the histopathological biopsy result. Commonly, patients present with vague symptoms and experience-prolonged delays in diagnosis with many inconclusive patient–physician contacts and resultant imaging leading to a high volume of false negative diagnoses. While osteoid osteoma is likely diagnosed on imaging alone, biopsy is the crucial step in the diagnostic workup for most primary benign spinal tumors [10,11,12,13,14,15]. Indications for biopsy, biopsy technique, harvesting success, and experience of histopathologists vary widely. Options for the final surgical strategy are versatile and include minimal-invasive techniques (e.g., radiofrequency ablation), intralesional curettage (i.e., Enneking-inappropriate (EI), or Enneking-appropriate (EA)) with or without subsequent stabilization, or even en bloc excision (Enneking-appropriate (EA)) of aggressive entities. Thus, it has been recommended by several authors to admit suspected primary tumor patients to specialized centers in order to achieve rapid diagnostic workup, sustainable biopsy results, correct indication, and ultimately appropriate intervention [16,17,18]. Satisfactory patients’ quality of life was thereby shown to be negatively associated with the onco-surgical outcome displayed by local recurrence and re-operation rates [19,20,21].

For that highly-heterogenous group of entities of extradural benign primary spinal tumors, the following study aims to provide descriptive data on the effect of different resection strategies on local recurrence and survival in patients younger than 25 years of age.

## 2. Materials and Methods

### 2.1. Study Design

This study was part of an international multicenter retrospective review of prospectively collected data or ambispective design with cross-sectional follow-up (PTRetro) performed by the AO Spine Knowledge Forum Tumor [22]. The study included 1495 patients that were surgically treated for primary spinal tumors between 1990 and 2012 by 13 tertiary spine centers in Europe and North America. Ethics approval was obtained at each participating center.

Patients were included in this analysis if they were less than or equal to 24.9 years of age at the time of diagnosis, had a follow-up greater than 6 months, and had a diagnosis of a primary benign spinal tumor (Figure 1). Data were captured and stored in a secure web-based program (REDCap, Vanderbilt University, Nashville, TN, USA).

Patient demographic data, symptoms, spinal tumor characteristics including pathology, Enneking classification, tumor size, and location were collected. The diagnostic and therapeutic approaches, surgical procedures, adjuvant therapies, local recurrence details, and cross-sectional survival were also recorded. Due to the higher aggressiveness and elevated recurrence rates, a subanalysis was performed for patient characteristics in cases with Enneking grades S1/S2 versus Enneking grade S3.

### 2.2. Treatment

Based on the heterogeneity of the population, a wide range of therapies were applied by the international centers because of the multidisciplinary decision-making with the local medical teams. All patients underwent surgical procedures. Tumor resection was performed either with or without spinal implant stabilization through single or combined approaches.

In moderate active tumors (S1, S2), intralesional surgeries are applied accordingly as Enneking appropriate resections (EA). In turn, EI constellations are usually not possible because clear margins have not been reached to treat these entities properly. In accordance with primary malignant tumors, S3 lesions need to undergo an extralesional resection to be EA. All intralesional procedures for S3 tumors are EI.

All patients that had a previous documented violation of the tumor prior to definitive treatment by biopsy or surgery were labeled as EI.

The surgical resection types ranged from palliative, intralesional subtotal, and gross total to en bloc resections.

### 2.3. Statistics

Patient data analysis was performed with descriptive statistics (mean and the standard deviation or median and percentiles for continuous variables and the absolute number and frequency distribution for categorical variables). Χ2 tests (Pearson and Fisher exact test) were used for categorical variables. Survival and local recurrences were illustrated by Kaplan–Meier curves. Significance was defined at a *p*-value of 0.05. The STATA software was used for statistical analyses (version 12.0, College Station, TX, USA).

## 3. Results

### 3.1. Patients

161 patients, 66 (41%) females and 95 (59%) males, with a mean age of 17.8 (±4.7) years at the time of the index intervention, underwent surgery between 1991 and 2011 in one of the centers. The median patient follow-up was 2.4 years. Patient demographic and clinical data are shown in Table 1, Table 2 and Table 3. Overall, 156 (96.9%) patients presented with pain at diagnosis and neurological deficits were found in 19% of cases. Eighteen patients (11.2%) presented with a pathologic fracture.

Diagnosis based on a biopsy at one of the study tertiary centers was achieved in 54.7% of cases. In 38.5% of cases, diagnosis was based on radiological imaging alone. Another 6.8% of patients received a biopsy in a hospital outside of the study centers and were thereby graded as EI. Table 1 and Table 2 show the summary of patient and tumor characteristics.

### 3.2. Treatment

Eighteen patients (11.2%) had previous surgery outside of one of the participating centers, wherein 12 (66.7%) had a reported intralesional tumor violation. All of these patients were documented as non-virgin tumors. To reduce intraoperative bleeding, 48 (31.6%) patients underwent preoperative embolization. The posterior approach was the most common (*n* = 126 (78.3%)), followed by combined (*n* = 24 (14.9%)), and anterior (*n* = 11 (6.8%)) approaches. Almost half of the patients (*n* = 75 (46.6%)) solely received resection of the tumor, while *n* = 86 (53%) needed an additional fixation for stabilization.

According to the histopathological reports from the surgical specimens, wide and marginal margins were achieved in 40 patients (26.3%), and intralesional margins in 112 patients (73.7%).

Additional adjuvant therapies (chemotherapy/radiation therapy) were mainly applied in cases with intralesional resections.

In all of the investigated patients, information about the Enneking appropriateness was available. One hundred patients (62.1%) received EA treatment, while 61 (37.9%) underwent EI treatment.

Additional adjuvant therapies received *n* = 10 (6.3%) patients, 5 (3.1%) received chemotherapy, and 9 patients (5.6%) underwent radiation therapy (EBRT).

Overall, the 10-year recurrence rate was 13.1%. For the three major entities osteoid osteoma (OO), aneurysmal bone cyst (ABC), and osteobolastoma (OBL), the local recurrence rates were 11.1%, 9.4%, and 11.3%, respectively. Out of these patients, no deaths were observed during the follow-up period. The majority of recurrences occurred in the first 2 years following surgery. There was no significant difference regarding time to first local recurrence by Enneking appropriateness (*p* = 0.415). Two of the patients died from tumor-related causes. See also Table 3

Overall, 69 cases with 16 GCTs and 53 OBL patients could be further sub-analysed and compared to the remaining benign lesions. According to Enneking, the majority (*n* = 53; 76.8%) of OBL/GCT entities were graded as aggressive (S3), while 23.2% were graded as active (S2). Table 4 shows the Enneking grading distribution for both entities. In the group of less aggressive entities (osteoid osteomas OO, aneurysmatic bone cysts ABC, schwannomas SCH, osteochondromas OCH, Langerhanns cell histiocytosis LCH; *n* = 91), 75% of patients underwent Enneking appropriate resections. In contrast, 44.9% of resections in the OBL/GCT group were Enneking appropriate. In less aggressive tumors, the 10-year local recurrence rate was 9.9%; all of them were diagnosed in the first 2 years following surgery (Figure 2a,b). A significant difference between EA and EI resections in terms of the time to first local recurrence was not found (*p* = 0.071). A local recurrence rate of 17.4% was found in in the OBL/GCT group, also presenting without significance between EA and EI resections (*p* = 0.501). While EI resections in the Kaplan–Meier analysis also showed early LR occurrence, EA resections showed recurrences over a longer period (see Figure 3a,b). Differences between OBL/GCT groups and less aggressive tumors were not significant (Table 4). The two patients that died in the overall cohort belonged to the OBL/GCT group.

## 4. Discussion

Extradural benign spinal tumors are rare and, due to the limited available literature, diagnostic and treatment algorithms do not exist. Even if the appearance of tumors is heterogeneous, diagnosis of a primary tumor is a dramatic occasion for each patient. In younger patients, tumor diagnosis may be even more devastating, and any necessary treatment can have a severe impact on further patients’ health-related quality of life. In some of the entities, tumor pathogenetic activity depends on the growth processes of the host and is thereby coupled to adolescence and younger age. Oncological algorithms and study protocols are established for children, teenagers, and young adults (TYA), with most of the schemes taking patients to their mid-twenties [3]. This study aimed to provide descriptive data on the effect of different resection strategies on local recurrence and survival in patients suffering from benign primary spinal tumors younger than 25 years of age.

The clinical characteristics of the 161 patients were quite comparable to the literature in terms of gender distribution, with a male preponderance and histopathological diagnosis headed by OBL, followed by OO, and ABCs [1,9,10,12,23,24]. The mobile spine was affected in the majority of patients, with a quarter of lesions spanning over more than two spinal levels (consequently graded as S3). The number one clinical symptom was pain. Only a relatively low number of patients presented with a pathological fracture or even a neurological deterioration. This constellation is typical and a mainstay for underestimation of the severity of the clinical problem, leading to, in most of the cases, insufficient diagnostic attempts and delayed late definite diagnosis [16] and possibility for consequent treatment.

As expected, patients treated surgically for benign primary tumors suffer from more aggressive types in accordance with the Enneking classification. While S1 tumors are usually treated due to the underlying pain, more active S2 tumors raise concerns regarding compartmental breach. Latter lesions represent only 22% of the OBL/GCT patient group in this study. Over 70% of S3 graded tumors are represented by these two entities and are thereby a known exception among benign tumors, characterized by crossing compartmental borders or even behaving similarly to malignancies [10,11,23,25]. In the presence of a histopathologically proven OBL or GCT, treatment concepts might have to be adapted to more aggressive treatment to reach adequate onco-surgical results [26].

However, estimation of the exact grading, especially to distinguish between active (S2) and aggressive (S3) lesions, seems to be quite problematic in terms of reliability and selectivity for tumor stage and extent [7]. Accordingly, multidisciplinary surgical planning has to additionally include tumor histological grade, stage and extent to determine the objective biological aggressiveness of the underlying tumor and in turn estimate the appropriateness of resection.

The key step to achieve the mentioned information (except for OO) is harvesting significant tissue volume by biopsy. Various aspects of biopsies prior to tumor surgery have been a matter of discussion [16,27] in terms of biopsy techniques, approaches, tracks, mass and representativeness of tissue, and necessity to conduct it at the center of index surgery. While delay of diagnosis is a known pre-hospital concern, a referral to experienced spine tumor centers is still vital. One fifth of young patients in this study showed a non-virgin tumor presentation to the tertiary center, known to be associated with higher local recurrence rates, decreased HRQOL, and even higher mortality rates in aggressive entities. In different publications [16,18,19], pooling of primary tumor patients at specialized centers has been recommended in the hope to increase local control rates and overall survival. In this cohort, 39% of lesions were diagnosed by radiological imaging only, which clearly exceeds the number of osteoid osteomas in an investigation where the diagnosis can be made without biopsy. This underlines still-existing differences and needs, even in experienced international centers, for diagnostic algorithms to achieve final diagnosis on standardized pathways.

Due to the heterogeneity of the investigated cohort of benign primary tumors, retrospective interpretation and judgement about obtained resection goals is difficult. It was previously mentioned that categorization of resection success by resection type is misleading and has to be orientated on the achieved histopathological margins. Outcome of surgical treatment is explained by the terms of Enneking appropriate (EA) or Enneking in-appropriate (EI) surgery instead [28]. For benign primary spinal tumors, an intralesional resection of an S2 tumor can be EA, while for an S3 graded tumor it is not considered sufficient [6,8]. In the overall cohort of this investigation, the results showed an overall 62% of EA resections according to the postoperative pathological reports. The high number of EI resections might have various reasons. Quite a high number of patients underwent previous biopsy or surgery (non-virgin) outside of one of the participating centers and were consequently graded as EI. The inclusion period for this study is quite long and radical more aggressive resection techniques were maybe incompletely accepted in the earlier phase. Another reason may be the avoidance to expose patients to the known elevated risks of radical more aggressive surgeries. Due to the close proximity of anatomical structures, elevated complication rates and postoperative physical impairment must be considered [23]. Furthermore, a discrepancy between planning and surgical realization might be a possible reason, especially with highly vascular, friable tumors where intraoperative fracture may be inevitable. This is a feature that would have been common with GCT prior to the introduction of neo-adjuvant Denosumab treatment [29].

Local recurrence as factor for an impaired overall outcome mostly occurred in the first 1–2 years following EI surgery, leading one to consider this group to be related to residual tumor growth vs. local recurrence per se. In the EA resection group, there was a delay until the time to the first local recurrence with a similar overall recurrence rate, however. This raises several questions regarding the dataset and is overall in contrast to our experience of similar pathologies in the PTRetro body of work.

Various limitations do exist for this analysis and careful interpretation of this data is recommended, especially in terms of a relatively short follow-up. Benign spinal lesions are usually not included in national registers, making standardized work-up even more difficult. In multicentric studies with retrospective inclusion, similar follow-up periods and length of follow-up do usually differ between the international centers, especially in patients with uneventful postoperative courses. Evaluation of outcome data might lead to wrong conclusions due to incoherent follow-up data.

Previous publications of this group and others have shown that margin-oriented resections for primary malignant spine tumors achieve low local recurrence rates and better overall outcome [19,20,21]. Similarly, these results have been shown with more biologically active and aggressive benign tumors [23,26]. This may be evident for cases that have breached compartments. For active intra-compartmental lesions, estimation of tumor cell proliferation by histopathological characterization and tumor imaging leave much more space for interpretation to estimate grading and necessary therapy. There might be greater challenges defining S3 vs. S2 tumors in this age group. However, we were not able to replicate the aforementioned results in a TYA cohort. A lack of clarity regarding grade, combined with a high rate of non-virgin tumors and perhaps a tendency to only operate in an EA manner for cases that are far more aggressive, may lead to a skew in the results.

With the growing availability of improved diagnostics, adjuvant therapies, and networks that deliver patients to specialist centers prior to tumor breach, it should be possible to determine decisive criteria or distinguish between aggressive and active tumors.

## 5. Conclusions

This is one of the largest international multicenter cohorts of surgically-treated benign spinal tumors in young patients. Compared to results previously published by this group, the evidently heterogenic younger patient cohort presented without a correlation between the grade of aggressiveness in resection and local recurrence rates at a midterm follow-up. Further prospective data are required to identify prognostic factors that determine oncological and functional outcomes for young patients suffering from these rare tumors.

## Figures and Tables

**Figure 1 cancers-15-00650-f001:**
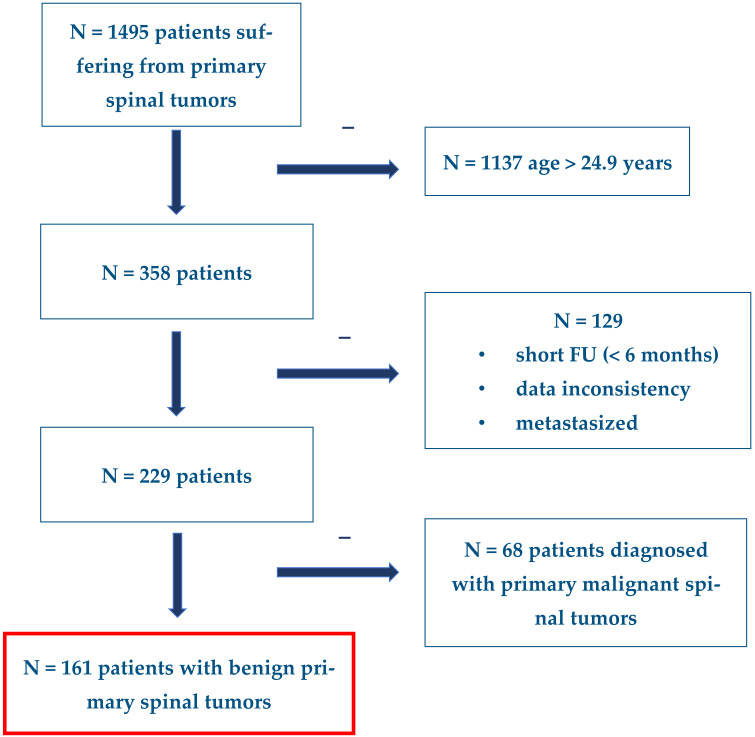
Flowchart of patient inclusion from the AO Spine Knowledge Forum’s PTRetro study.

**Figure 2 cancers-15-00650-f002:**
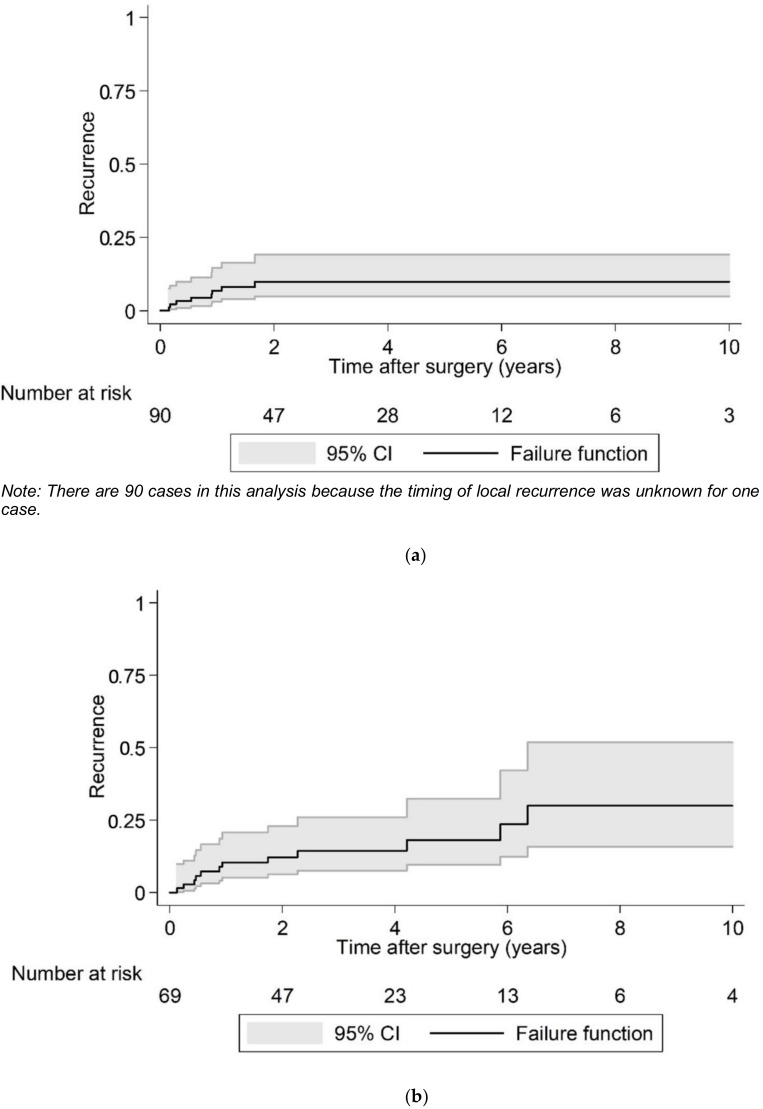
Kaplan–Meier curves of time to first local recurrence following surgery for less aggressive entities (**a**) and GCT/OBL patients (**b**).

**Figure 3 cancers-15-00650-f003:**
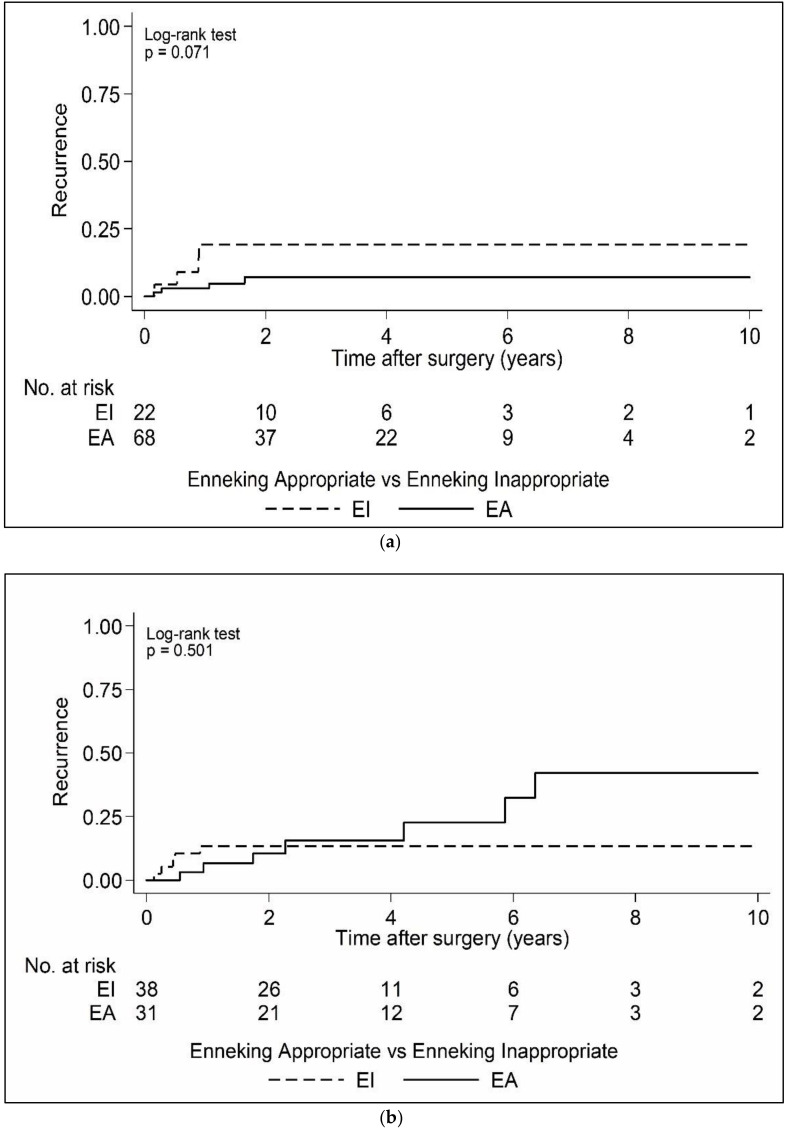
Kaplan–Meier curves of time to first local recurrence following surgery by Enneking appropriateness for less aggressive entities (**a**) and GCT/OBL patients (**b**).

**Table 1 cancers-15-00650-t001:** Summary of patient demographic and clinical characteristics. Abbreviations are giant cell tumor (GCT), osteoid osteoma (OO), osteoblastoma (OBL), osteochondroma (OCH), aneurysmal bone cyst (ABC), Langerhanns cell histiocytosis (LCH), schwannoma (SCH).

Variable		*n* (%)
Diagnosis (*n* = 161)	GCT	16 (9.9)
	OO	45 (28.0)
	OBL	53 (32.9)
	OCH	4 (2.5)
	ABC	32 (19.9)
	LCH	2 (1.2)
	SCH	9 (5.6)
Gender (*n* = 161)	Female	66 (41.0)
	Male	95 (59.0)
Age at time of diagnosis (years) (*n* = 161)		17.0 ± 4.7
Age at time of surgery (years) (*n* = 161)		17.8 ± 5.3
Pain at Diagnosis (*n* = 161)	No	5 (3.1)
	Yes	156 (96.9)
Pathologic Fracture at Diagnosis (*n* = 161)	No	143 (88.8)
	Yes	18 (11.2)
Previous Spine Tumor Operation (*n* = 161)	No	143 (88.8)
	Yes	18 (11.2)
Preoperative Frankel and ASIA Score * (*n* = 158)	A	2 (1.3)
	B	0 (0.0)
	C	6 (3.8)
	D	22 (13.9)
	E	128 (81.0)
Local recurrence over 10 years postoperative (*n* = 160)	No	139 (86.9)
	Yes	21 (13.1)
Survival over 10 years postoperative (*n* = 160)	Alive	158 (98.8)
	Dead	2 (1.3)

Data are presented as *n* (%) or Mean ± Standard Deviation. * When a discrepancy between ASIA and Frankel score occurred, the more severe score was chosen.

**Table 2 cancers-15-00650-t002:** Summary of tumor characteristics.

Variable		*n* (%)
Tumor Size (cm)	Tumor Volume Ellipsoid Body (cm^3^) * (*n* = 132)	4.2 (1.0, 9.4)
	<5	81 (61.4)
	≥5	51 (38.6)
Spinal level (*n* = 161)	Mobile	142 (88.2)
	Fixed	19 (11.8)
Level by Cervical, Thoracic, Lumbar, Sacral (*n* = 152)	Cervical	48 (31.6)
	Thoracic	49 (32.2)
	Lumbar	42 (27.6)
	Sacral	13 (8.6)
Number of Vertebral Levels Spanned by the Tumor (*n* = 161)	1	121 (75.2)
	≥2	40 (24.8)
Tumor Grade (Enneking Classification) (*n* = 161)	S1	14 (8.7)
	S2	73 (45.3)
	S3	74 (46.0)

Data are presented as *n* (%) or Median (p25, p75); * Tumor Volume Ellipsoid Body = π/6 × height × width × depth.

**Table 3 cancers-15-00650-t003:** Summary of treatment details.

**Variable**		***n* (%)**
Preoperative Embolization (*n* = 152)	No	104 (68.4)
	Yes	48 (31.6)
Surgical Approach (*n* = 161)	Anterior	11 (6.8)
	Posterior	126 (78.3)
	Anterior/Posterior	4 (2.5)
	Posterior/Anterior	17 (10.6)
	Posterior/Anterior/Posterior	1 (0.6)
	Other	2 (1.2)
Fixation Used (*n* = 161)	Anterior	5 (3.1)
	Posterior	72 (44.7)
	Both	9 (5.6)
	None	75 (46.6)
Neurology Sacrificed: Cord (*n* = 159)	No	159 (100.0)
	Yes	0 (0.0)
Neurology Sacrificed: Cauda Equina (*n* = 159)	No	159 (100.0)
	Yes	0 (0.0)
Neurology Sacrificed: Nerve Roots (*n* = 159)	No	144 (90.6)
	Yes	15 (9.4)
Pathology result from the surgical specimen (*n* = 152)	Wide or marginal	40 (26.3)
	Intralesional	112 (73.7)
Enneking appropriateness (*n* = 161)	EA	100 (62.1)
	EI	61 (37.9)
Adjuvant therapy (*n* = 160)	No	150 (93.8)
	Yes	10 (6.3)
Timing of chemotherapy (*n* = 161)	Preop	0 (0.0)
	Postop	4 (2.5)
	Both	0 (0.0)
	Neither (no chemo)	156 (96.9)
	Timing unknown	1 (0.6)
Timing of radiation therapy (*n* = 161)	Preop	0 (0.0)
	Postop	8 (5.0)
	Both	0 (0.0)
	Neither (no radiation)	152 (94.4)
	Timing unknown	1 (0.6)
Type of Radiation Therapy given (*n* = 9)	Conventional	7 (77.8)
	IMRT	0 (0.0)
	Radiosurgery	0 (0.0)
	Proton Beam	0 (0.0)
	Unknown	2 (22.2)

**Table 4 cancers-15-00650-t004:** Comparison of patient characteristics between less aggressive entities (left side) and GCT/OBL patients (right side).

Variable	*n* (%)	Variable	*n* (%)	*p*
**Diagnosis**				
**(*n* = 92)**		**(*n* = 69)**		
Osteoid osteoma	45 (48.9)	Giant cell tumor	16 (23.2)	
Osteochondroma	4 (4.3)	Osteoblastoma	53 (76.8)	
ABC	32 (34.8)	**Tumor grade (*n* = 69)**		
LCH	2 (2.2)	S2	16 (23.2)	
Schwannoma	9 (9.8)	S3	53 (76.8)	
**Enneking appropriateness**				
**(*n* = 92)**		**(*n* = 69)**		
EA	69 (75.0)	EA	31 (44.9)	>0.05
EI	23 (25.0)	EI	38 (55.1)	
**Local recurrence over 10 years postoperative**				
**(*n* = 91)**		**(*n* = 69)**		
No	82 (90.1)	No	57 (82.6)	
Yes	9 (9.9)	Yes	12 (17.4)	>0.05
**Survival over 10 years postoperative**				
**(*n* = 91)**		**(*n* = 69)**		
Alive	91 (100.0)	Alive	67 (97.1)	
Dead	0 (0.0)	Dead	2 (2.9)	>0.05

## Data Availability

Data Management and availability.

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
