# Peer review of "Outcomes of Surgical Treatment for Extradural Benign Primary Spinal Tumors in Patients Younger than 25 Years: An Ambispective International Multicenter Study"

_cancers, 2023, doi:10.3390/cancers15030650_

Round 1

Reviewer 1 Report

The authors should mention in the title that this review is about spinal bony tumor, because they didn't include spinal cord tumor.

at the beginning of the discussion part: The first paragraph, Im not sure if the authors left this on purpose.

Would ask the authors to mention why they chose the 25 of age as cut off in the introduction

Manuscript very easy and smooth to read, 

Analysis is very well done and the tables are very easy to grasp, 

All in all manuscript is very well written

Author Response

Response to Reviewer 1

comment No. 1

The authors should mention in the title that this review is about spinal bony tumor, because they didn't include spinal cord tumor.

The reviewer is right. We changed the title to: „Outcomes of surgical treatment for extradural benign primary spinal tumors in patients younger than 25 years: an ambispective international multicenter study“.

Additionally, we added the term “extradural” to different sentences in the manuscript (Simple Summary, Abstract, Introduction, Discussion).

comment No. 2

At the beginning of the discussion part: The first paragraph, I´m not sure if the authors left this on purpose

The reviewer is right at this point, too. We removed the mentioned paragraph accordingly.

comment No. 3

I would ask the authors to mention why they chose the 25y of age as a cut off on the introduction

The reviewer is right at this point, too. We re-arranged the existing paragraph to highlight that point as follows: “Oncological approaches for primary spinal tumors in younger patients often significantly differ from adult treatment regimens. In the common literature they are thereby divided in two age groups: from birth until 14 years of age and from 15 to 25 years of age [3,9]. However, multidisciplinary adjuvant treatment often recommends similar treatment protocols for the same entities for all the patients under 25 years of age [1]. While international coordinated adjuvant treatment algorithms and study protocols are used, knowledge about onco-surgical outcomes of extradural benign primary spinal tumors in young patients is limited. Neither unique resection strategies (e.g. Enneking Inappropriate (EI) vs. Enneking Appropriate (EA)) nor clear-cut reconstruction guidelines exist.“

comment No. 4

Manuscript very easy and smooth to read, analysis is very well done and the tables are very easy to grasp. All in all manuscript is very well written.

Thank you very much for your kind review.

Reviewer 2 Report

The authors have submitted an original article that retrospectively reviewed primary benign spinal tumors in patients younger than 25 years of age. This international multicenter study on a relatively rare disease entity provides descriptive data on the surgical outcome of primary benign spinal tumors. As the main result, the authors found that the aggressiveness of resection based on Enneking appropriateness is not associated with local recurrence. However, the study has significant flaws to be considered as a publication. 

First, the study cohort is too heterogeneous. Primary benign spinal tumors significantly vary in their behavior and natural course. For example, giant cell tumors can metastasize to vital organs and affect a patient's survival, whereas osteoid osteomas are mostly indolent. Although the authors have provided subgroup analyses on S3 tumors in the manuscript, there seems to be only limited value for presenting surgical outcomes of this heterogeneous group of tumors. Also, even considering the rarity of these primary benign spinal tumors, the total sample size seems to be too small for an international multicenter study. 

Second, the follow-up period for the total cohort seems to be too short given the study period. The authors stated that the patients were enrolled between 1990 and 2012. However, the median follow-up period is only 2.4 years. Can the authors provide the reason for such a short follow-up period? In the inclusion criteria and Figure 1, they only excluded patients with a follow-up period of 6 months or less. This is too short for studies on survival or local recurrence of a tumorous condition. 

For these reasons, the conclusion of this study that the aggressiveness of resection based on Enneking appropriateness is not associated with local recurrence cannot be accepted. A longer follow-up period in a larger sample is required for meaningful results. 

In addition, the overall quality of the manuscript requires significant improvements. The followings are a few details of other concerns and issues regarding the study. 

#. In line 46, the number of patients with local recurrence is omitted. 

#. Please consider using the lowercase letter n when describing the number of patients. The uppercase letter is overused. For instance, in lines 43-44, "Tumor-related pain was present in N=156(96.9%) patients" should be changed to "Tumor-related pain was present in 156(96.9%) patients"

#. Line 125-128 is hard to understand. 

#. "As already described for benign spinal tumors, N=156 (96.9%) 141 of patients presented with pain at diagnosis and only N=18 (11.2%) with a pathologic fracture" Where did you already describe? Pathologic fracture is not a symptom and should not be described here. 

#. In line 143, what does "5.1% A-C Frankel and ASIA Score" mean? 

#. Line 145-149 is so confusing. 

#. Please provide abbreviations in Table 1. The last row in Table 1 should be deleted. 

#. ASIA score, which has been developed for spinal cord injury, is often inappropriate to describe the neurological status of spinal tumor patients.  

#. In Table 1, "*When a discrepancy between ASIA and Frankel score occurred, the more severe score was chosen." I cannot understand how you can merge these two totally independent systems (ASIA and Frankel) to evaluate a patient's functional status.  

#. In line 171, 62% cannot be referred to as the majority. 

#. How did you diagnose the local recurrence? The credibility of the 10-year local recurrence rate estimated by the Kaplan-Meier curve is low when the median follow-up period is only 2.4 years. In line 175, how can you say that the majority of recurrences occurred in the first 2 years following surgery, when the total cohort has such a short follow-up period?

#. How did you differentiate between the local recurrence and the progression of residual tumor in Enneking inappropriate resection patients?

Author Response

Response to Reviewer 2

We are grateful for the input from Reviewer 2 and will address the two main criticisms.

Whilst we agree that there are some variances in outcomes seen between tumour pathologies/groups there are obviously many similarities between the cohorts that we have chosen. These similarities have driven the original Enneking Classification and the subsequent recommendations regarding surgery.

The dates incorporating the study date back to a period when centralization of tumour services was more unusual than it is now.  It is certainly possible that regional and supraregional shifts in referral pattern during the study period explains somewhat the bias towards the shorter FU that we see. That said we have also shown that for these tumours, recurrence is seen within the first two years. Given that many of these tumours are often highly symptomatic, early discharge of pain free patients with certain pathologies would also be commonplace and this would be particularly the case for osteoid osteoma 45/161. This would also foreshorten the expected follow up statistics. Despite these shortcomings we believe this paper represents a significant contribution to the literature across the breadth of these spinal tumours. We acknowledge that a prospective multicentre study will provide even better insight. We will endeavour to deliver this in due course.

comment No. 1

First, the study cohort is too heterogenous. Primary benign spinal tumors significantly vary in their behaviour and natural course…Also even considering the rarity of these primary benign spinal tumors, the total sample size seems to be too small for an international multicenter study.  

Thank you very much for your comment, that absolutely highlights the general problem for those rare entities. The mentioned PTRetro database includes 1495 patients surgically treated for extradural primary spinal tumors. To our knowledge this is still one of the largest ambispective databases in the literature reflecting that only little data for younger patient might be available even in multicentric approaches. Aim of the presented analysis was to summarize data about patients under 25 years of age. Their treatment significantly differs from adult patients from a medical point of view, but no algorithms do exist for standardized diagnostic approaches and treatments. As with malignant spinal sarcoma, primary benign extradural tumors are highly heterogenous by origin, biological behaviour, histopathological and radiological appearance. However, all of them are classified using the Enneking classification system (originally published for extremities and transferred to spinal tumors) and corresponding treatment recommendations do exist. The aim of the current study was to detect possible inconsistencies of the common classification, treatment and the outcome of this cohort. To address that point, we modified the following sentences to the Introduction section:

“According to the widely accepted Enneking staging system for primary bone tumors [6-8], benign entities significantly vary from being classified as "latent (S1)" with slow intracompartmental growth, "active (S2)" defined as clinically symptomatic but still intracompartmental, and "aggressive (S3)", characterized as non-respecting of compartmental borders, and as an extreme, present as tumors with malignant-like behaviour including the ability to metastasize [6,8].”

“For that highly heterogenous group of entities of extradural benign primary spinal tumors, the following study aims to provide descriptive data on the effect of different resection strategies on local recurrence and survival in subjects younger than 25 years of age.” 

comment No. 2

The follow-up period for the total cohort seems to be too short given the study period…Can the authors provide the reason for such a short follow-up period? The authors excluded patients with a follow-up period of 6 months or less. This is too short for studies on survival.

We absolutely agree that this is an important weakness of the study. Benign spinal lesions are usually not included to national registers making standardized work-up even more difficult. Postoperative uneventful courses without complications (as usually seen in osteoid osteoma that do represent about a quarter of the cohort) are usually not regularly followed-up. In multicentric studies with retrospective inclusion, similar follow-up periods and length of follow-up do differ between the international centers. From a historical point of view referral to and centralisation in tertial centers was not common in the early years of that study and is still not common standard internationally. This might lead to irregular follow-up periods in that era. Therefore, the inclusion cut-off point with 6 months was chosen to include cases even with short term follow-up. However, we agree that only limited conclusions can be drawn regarding the long-term outcome of the cases.

To underline that point, we added the following paragraph to the Discussion section: “Various limitations do exist for this analysis and careful interpretation of this data is recommended, especially in terms of relatively short follow-up. Benign spinal lesions are usually not included to national registers making standardized work-up even more difficult. In multicentric studies with retrospective inclusion, similar follow-up periods and length of follow-up do usually differ between the international centers, especially in patients with uneventful postoperative courses. Evaluation of outome data might lead to wrong conclusions due to incoherent follow-up data.”

comment No. 3     

In line 46, the number of patients with local recurrence is omitted

The reviewer is right at this point, too. The manuscript was thoroughly revised. The number omitted was added.

comment No. 4

Please consider using the lowercase letter n. The uppercase letter is overused.

Accordingly, we changed the manuscript text when applicable.

comment No. 5

Line 125-128 is hard to understand

We absolutely agree and replaced that sentence.

comment No. 6

“As already described for benign spinal tumors…” Where did you already describe? Pathologic fracture is not symptom and should not be described here.

The reviewer might be right that this could irritate some readers. The authors tried to show parallels to the common literature and the most frequent presentations of benign spinal tumors. Even if a fracture is not a symptom, it is sometimes a first sign of the disease. The text was changed as follows:

“Overall, 156(96.9%) patients presented with pain at diagnosis and neurological deficits were found in 19% of cases. Eighteen patients (11.2%) presented with a pathologic fracture.” 

comment No. 7

  • In line 143, what does “5.1% A-C Frankel and ASIA Score” mean?
  • ASIA Score, which has been developed for spinal cord injury, is often inappropriate to describe the neurological status of spinal tumor patients
  • In table1, “when a discrepancy between ASIA and Frankel score occurred, the more severe score was chosen.”

The reviewer is right in this point and that might be irritating. The Frankel classification of spinal injuries was popular from the 70s on of the last century and widely used with the well-known five categories. At the beginning of the 90s it was replaced by the ASIA Impairment Scale that closely follows the Frankel scale but at the same time differs in several important respects defining more clearly and unambiguous by including information of the sacral segments. From a scientific point of view that change improved reliability and is more predictive of prognosis. Even if they were designed for spinal cord injury, both are widely used throughout the literature especially in retrospective analysis about epidural tumor involvement and their use increase comparability. Another reason of their use is the simple fact of availability, because they are usually implemented in patients` diagnostic work up. Most of the data of the presented study was gathered retrospectively, some centers used Frankel scale (n=138) in the earlier inclusion period, while ASIA was the standard at later time points (n=123) or even both were used. For prospective work of this group, more tumor related scores are already implemented. According to the authors view it is acceptable in that retrospective cohort to give an impression and overview of the severity of neurological deficits if both information’s are included.

The mentioned sentence was modified.

comment no. 8

Line 145-149 is so confusing.

To make this paragraph clearer, it was re-arranged as follows:

“Diagnosis based on a biopsy performed at one of the tertiary study centers was achieved in 54.7% of cases. In 38.5% of cases diagnosis was based on radiological imaging alone. Another 6.8% of patients received a biopsy in a hospital outside of the study centers and where thereby graded as EI.”

comment No. 9

Please provide abbreviations in Table 1. The last row in table 1 should be deleted.

We totally agree with the reviewer at this point. We added the abbreviations accordingly. Please see also the reply on comment no. 7

comment no. 10

In line 171, 62% cannot be referred to as the majority.

To avoid ambiguity, we changed the sentence as follows: “One hundred patients (62.1%) received EA treatment, while 61(37.9%) underwent EI treatment.”

Comment no. 11

How did you diagnose the local recurrence? The credibility of the 10-year local recurrence rate estimated by the Kaplan-Meier curve is low when the median follow-up period is only 2.4 years. In line 175, how can you say that the majority of recurrences occurred in the first 2 years following surgery, when the total cohort has such a short follow-up period?

As we stated at the beginning, we do agree that there might be a bias due to shorter follow-up periods in that study. However, we were able to show that local recurrence occurs even in benign entities with a midterm follow-up in the first two years. As the mentioned sentence might be wrongly interpretated by the readers, we removed that sentence. 

Comment no. 12

How did you differentiate between the local recurrence and the progression of residual tumor in Enneking inappropriate resection patients?

As by definition, Enneking inappropriate resections leave tumor tissue to some extent at the surgical field. Not self-limiting progression will thereby lead to a local recurrence. In accordance with the previous surgical resection type, new diagnostic imaging was analyzed in comparison to pre-operative ones to set tumor growth.    

Round 2

Reviewer 2 Report

I appreciate the authors' effort in revising and improving the manuscript thoroughly. However, I still think this study provides only limited value because the fundamental issues, including the heterogeneity of the study cohort, the small sample size of individual tumors, and too short follow-up periods, could not be solved in the revised manuscript.

Author Response

Dear Reviewer 2,   Thank you again for your comments and for giving us the chance to re-revise the manuscript. Your comments were really helpful for further developing the manuscript. We do understand your concerns. But we still believe this paper represents a significant contribution to the literature across the breadth of these spinal tumors.   Thank you once again for your contributions   Kind regards